# Covalent penicillin-protein conjugates elicit anti-drug antibodies that are clonally and functionally restricted

Lachlan P. Deimel [1,9] ✉, Lucile Moynié[2], Guoxuan Sun[2], Viliyana Lewis [2], Abigail Turner [2], Charles J. Buchanan [2,3,4], Sean A. Burnap[4,5], Mikhail Kutuzov[1], Carolin M. Kobras[1], Yana Demyanenko [2,6], Shabaz Mohammed [2,3,5], Mathew Stracy[1], Weston B. Struwe [4,5], Andrew J. Baldwin [2,3,4], James Naismith [2], Benjamin G. Davis [2,3,6] ✉ & Quentin J. Sattentau [1,7,8] ✉

Many archetypal and emerging classes of small-molecule therapeutics form covalent protein adducts. In vivo, both the resulting conjugates and their off-target side-conjugates have the potential to elicit antibodies, with implications for allergy and drug sequestration. Although β-lactam antibiotics are a drug class long associated with these immunological phenomena, the molecular underpinnings of off-target drug-protein conjugation and consequent drug-specific immune responses remain incomplete. Here, using the classical β-lactam penicillin G (PenG), we probe the B and T cell determinants of drug-specific IgG responses to such conjugates in mice. Deep B cell clonotyping reveals a dominant murine clonal antibody class encompassing phylogenetically-related *IGHV1*, *IGHV5* and *IGHV10* subgroup gene segments. Protein NMR and x-ray structural analyses reveal that these drive structurally convergent binding modes in adduct-specific antibody clones. Their common primary recognition mechanisms of the penicillin side-chain moiety (phenylacetamide in PenG)—regardless of CDRH3 length—limits cross-reactivity against other β-lactam antibiotics. This immunogenetics-guided discovery of the limited binding solutions available to antibodies against side products of an archetypal covalent inhibitor now suggests future potential strategies for the 'germline-guided reverse engineering' of such drugs away from unwanted immune responses.

In isolation, non-protein, low molecular weight compounds are typically non-immunogenic to the mammalian immune system. As exemplified by classical hapten-carrier biology, antibody responses against small molecules such as 4-hydroxy-3-nitrophenol acetyl (NP) require conjugation to a suitable carrier protein[1]. However, upon attachment to protein, resulting epitopes arise within an antigenic complex that may cross-link cognate B cell receptors (BCRs) and that are associated with peptidic components that may be presented to T helper (Th) cells;

[1]Sir William Dunn School of Pathology, University of Oxford, Oxford OX1 3RE, UK. [2]Rosalind Franklin Institute, Harwell Science and Innovation Campus, Oxford OX11 0FA, UK. [3]Department of Chemistry, University of Oxford, Oxford OX1 3TA, UK. [4]Kavli Institute for Nanoscience Discovery, Dorothy Crowfoot Hodgkin Building, University of Oxford, Oxford OX1 3QU, UK. [5]Department of Biochemistry, University of Oxford, Oxford OX1 3QU, UK. [6]Department of Pharmacology, University of Oxford, Oxford OX1 3QT, UK. [7]The Max Delbrück Centre for Molecular Medicine, Campus Berlin-Buch, 13125 Berlin, Germany. [8]Experimental and Clinical Research Center (ECRC), Charité Universitätsmedizin Berlin and Max-Delbrück-Center for Molecular Medicine, Lindenberger Weg 80, 13125 Berlin, Germany. [9]Present address: Laboratory of Molecular Immunology, The Rockefeller University, New York, NY 10065, USA. ✉e-mail: Lachlan.Deimel@path.ox.ac.uk; Ben.Davis@chem.ox.ac.uk; Quentin.Sattentau@path.ox.ac.uk

together, these can impart combined help to propagate a specific B cell population to a given attached compound[2–4].

In principle, these mechanisms may extend to small-molecule drugs, particularly those with reactive functional groups that drive covalent conjugation with endogenous proteins under physiological conditions[5,6]. Unwanted immune responses to covalent bond-forming drugs, particularly in the form of anti-drug antibody (ADA) responses, include hypersensitivity and allergy reactions, the most severe of which can be life-threatening. Whilst the number of covalent bond-forming drugs (e.g. covalent inhibitors) in clinical use has been traditionally limited[7], in recent years there has been strong renewed interest[8,9] yet notably little analysis of unwanted drug-immune system interactions.

The best-characterised examples of unwanted immunogenicity from drug-protein conjugates are β-lactam antibiotics, such as penicillin G (PenG)[10]. As the electrophilic source of its inhibitory activity, the β-lactam group of PenG may also drive background / side reactivity with off-target biological nucleophiles leading to protein conjugation via primary amine-containing sidechains of lysine (and potentially other nucleophilic residues including arginine, histidine, and cysteine), as has been observed under some buffer conditions[11,12]. Such protein-PenG complexes are the antigenic determinants of antibiotic hypersensitivity. The mechanistic underpinnings of the hypersensitivity reaction are immunologically heterologous, with the most common and well characterised being T helper (Th) cell-mediated (type IV) that may be elicited in up to 30% of the population[13–17]. However, the most clinically severe forms of drug hypersensitivity are antibody-mediated, particularly IgE-induced anaphylaxis[18]. IgG-mediated hypersensitivity is less severe but relatively common[18,19].

Penicillin is one of the most frequent causes of anaphylaxis and anaphylaxis-related deaths in humans[19]. However, penicillin allergy diagnosis is currently highly inaccurate. Nearly 6% of the general population in the UK are recorded as having a penicillin allergy, yet more than 95% of these patients can ultimately tolerate this class of drug, indicating that most patients are falsely recorded as allergic[20]. Patients with a penicillin allergy record have an increased risk of *Clostridioides difficile* and Methicillin-resistant *Staphylococcus aureus* infections and death; this is presumably through increased use of alternatives to β-lactam antibiotics[21]. Furthermore, penicillin allergy diagnosis is associated with higher numbers of total antibiotic prescriptions[22], undermining antimicrobial stewardship goals and increasing the risk for antimicrobial resistance[23]. A better understanding of the immunological basis of penicillin hypersensitivity is therefore vitally needed to help predict which antibiotic recipients are, or will become, allergic[24,25], and to inform potential future deleterious immune reactions against new generations of covalent bond-forming drugs.

Notably, although the first descriptions of penicilloyl-directed serological responses were made in 1961[12], key phenomena remain incompletely understood, including (i) the biochemical basis of PenG–protein adduction in vivo and in vitro; (ii) the relative immunogenicity of fully chemically characterised and purified PenG adducts; (iii) the immunophylogenetics of B cells specific to PenG-protein complexes; and (iv) the structure/function characteristics of antibody clones specific to these adducts.

Here, through systematic complementary biochemical, structural and clonotypic analyses of the relationship between the protein-conjugating properties of PenG and its immunogenicity in a mouse model, we now fully map the PenG-specific antibody response. We find that the ADA response is based upon a restricted cluster of highly related B cell germline clonal families that, regardless of CDRH3 length, engage the penicillin sidechain via conserved binding modes. These findings offer a rational basis for understanding ADA responses, and suggest that antibodies have limited binding solutions that in turn may inform drug 'reverse engineering' to avoid ADA.

## Results

### Immunogenicity of 'pre-complexed' penicillin-protein conjugate antigen

PenG has constituent β-lactam, thiazolidine and phenylacetamide sidechain moieties (Fig. 1a); the β-lactam ring is long known to react with nucleophilic protein sidechains, including the off-target ε-amino groups of lysine residues leading to the formation of subsequent ε-amide-drug adduct forms such as those formed via β-lactam ring opening[3,11,12,26–28] (Fig. 1b). To probe PenG-protein off-target conjugation, we first titrated adduct formation on the model protein hen egg lysozyme (HEL, ~14 kDa, 129 a.a.), chosen for its stability and relatively evenly distributed (six) lysine (Lys) residues. Various buffer, drug amounts and pH conditions were tested in vitro, and global site-specific drug occupancy was evaluated via mass spectrometry (MS). These revealed pH- and buffer-modulated conjugation levels (Fig S1a; Document S1). Mapping of adduct formation through tryptic digest followed by site-specific liquid chromatography tandem MS (LC-MS/MS) analysis confirmed adduct formation at all lysine residues 1, 13, 33, 96/97 and 116 (Fig S1b). These reaction data informed ex vivo conjugation of PenG to various recombinant carrier proteins at close-to-physiological pH for subsequent immunisation.

Pilot immunogenicity analysis of HEL-PenG conjugates formulated in aluminium hydroxide (alum) adjuvant was conducted by subcutaneous (s.c.) administration to C57BL/6 mice. Antisera were titrated by ELISA on an unrelated PenG-modified protein (human serum albumin-PenG; HSA-PenG) to determine the titres specifically against the penicilloyl adduct. This revealed modest but significant ($P < 0.05$ Mann–Whitney U) isotype-switched IgG responses raised against the drug adduct (Fig S1d–g).

Although a useful model antigen for biochemical characterisation and pilot immunogenicity analysis, HEL is a weak Th cell antigen in mice[29,30]. We therefore next evaluated the antibody response generated by PenG pre-complexed to more antigenic HSA using the conditions optimised for HEL (Fig. 1c). Site-specific occupancy of penicilloyl adducts was evaluated via LC-MS/MS and again diverse lysine occupancy was observed (Fig S2). These occupancy data are concordant with previously published drug modification sites of HSA[28,31–33]. Mice were immunised with HSA or HSA-PenG formulated in alum, followed by periodic blood sampling (Fig. 1d). Anti-penicilloyl serum IgG responses were measured by ELISA against HEL-PenG; no IgG cross-reactivity was detected against unmodified HEL (Fig. 1e). However, strikingly, post-prime HSA-PenG antisera displayed considerable IgG reactivity with HEL-PenG, whereas no reactivity was detected in the HSA-alone antiserum (Fig. 1f). Together, these immediately suggested an anti-PenG-adduct-specific response. Post-boost and at the terminal timepoint, the median HEL-PenG-specific IgG endpoint titre (EPT) was marked at ~$2.2 \times 10^5$ for the HSA-PenG antisera and a near-baseline EPT of ~$1.7 \times 10^2$ for the control HSA antisera ($P = 0.029$, Mann-Whitney test) (Fig. 1g). Analysis of CD4+ Th responses revealed significant T cell proliferation and IFN-γ production only in the HSA-PenG-restimulated cells, consistent with the adducted protein being most efficiently captured and processed by B cells and presented to Th cells (Fig S3).

The observed antibody responses were generated against a protein-conjugated PenG derivative. Characterisation was consistent with direct β-lactam opening, but we cannot discount a pathway involving intermediates of penicillanic acid (Document S1)[27,28]. HSA-PenG antiserum binding to HEL-PenG was out-competed by free PenG, with a median IC$_{50}$ of 1.8 mM, while kanamycin (an unrelated non-β-lactam-type antibiotic) failed to detectably compete for antibody binding (Fig. 1h). These data reveal that antibodies raised against the autologous PenG adduct are cross-reactive with free PenG, suggesting recognition of a common motif that is not the

'opened' β-lactam. Administration of HSA-PenG did not affect the antibody responses against the protein unmodified protein backbone, compared with mice immunised with HSA alone ($P > 0.9999$, Mann–Whitney test) (Fig. 1i,j).

## Self-protein carriers elicit penicillin-specific antibodies

Having demonstrated strong B cell immunogenicity of PenG conjugated to foreign proteins (HEL or HSA), we next tested a native and otherwise tolerogenic self-antigen, mouse serum albumin (MSA), as a more physiologically relevant model[34]. First, pure MSA was precomplexed with PenG in buffer under the optimised conditions, as described previously, and the resultant MSA-PenG was analysed via LC-MS/MS (Fig S4). Animals were immunised three times (wk 0, 4 and 8) with or without alum and terminal (wk 10) IgG antibody titres were evaluated by ELISA against OVA-PenG (Fig. 2a). 5/6 (83%) animals immunised with MSA-PenG in alum elicited a detectable serum endpoint titre against the adduct (median titre of ~ $5.2 \times 10^3$) and 3/8 (38%) responders in those immunised with MSA-PenG even without adjuvant (Fig. 2b). Surprisingly, these data show that extrinsic adjuvantation is not required to elicit anti-adduct IgG responses even when presented on a modified self-protein. By contrast, no animal immunised with unmodified MSA alone with or without alum gave a detectable PenG-specific

response ($P = 0.033$, Kruskal-Wallis test) (Fig. 2b). Although the MSA-PenG-elicited PenG-specific antibody titres were lower compared with HSA-PenG (Figs. 1a, 2b), these data show that a penicillin-specific IgG response can be generated even when using an otherwise highly T-immunorecessive 'self-derived' backbone, and that self-proteins can act as 'non-self' immunogenic carriers when modified by drug. No reactivity was detected against OVA in any group (Fig. 2c).

Finally, to develop an ex vivo model for self-protein:PenG adduct formation and immunogenicity, we tested the ability of complete mouse serum itself to act as a carrier for PenG. PenG-naïve mice were bled, serum isolated and incubated ex vivo with 2 mg/mL PenG for 16 h. This concentration was chosen to mimic a typical serum concentration in humans[35]. The resulting serum-adduct mixture was then administered intravenously (i.v.), into the autologous mice, with and without alum, to mimic a clinical route of penicillin administration (Fig. 2d). Terminal IgG endpoint titres were evaluated against OVA-PenG, which revealed 6/8 (75%) of animals primed and boosted with adjuvanted serum-PenG responded with a detectable IgG titre against OVA-PenG and a median titre of ~ $2.5 \times 10^3$ (Fig. 2e,f). Interestingly, only a single animal that received unadjuvanted serum-PenG gave a detectable PenG-specific IgG titre, implying that adjuvantation, such as might be generated by bacterial infection during therapeutic penicillin

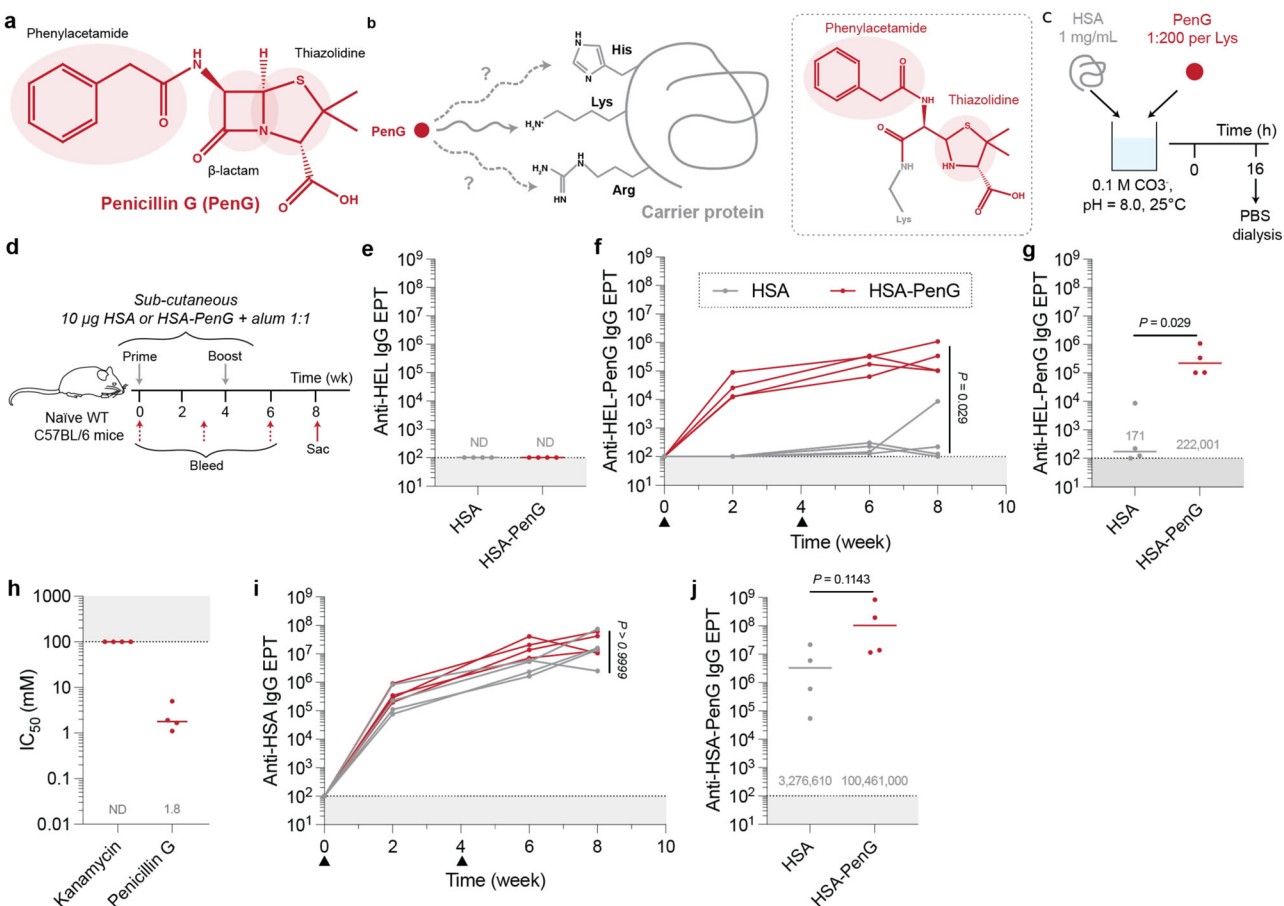

**Fig. 1 | Serological responses against PenG-derived penicilloyl protein adducts.** **a** Chemical structure of PenG. **b** Proposed target residues of electrophilic β-lactam; primary reported target of lysine with potential targeting of arginine and histidine. **c** 1 mg/mL HSA and PenG (1:200 per Lys) were mixed in vitro with 0.1 M HCO₃⁻, pH = 8.0. This was left at 25 °C for 16 h before dialysis into PBS. **d** Sex-matched 6-week-old naïve WT C57BL/6 mice were twice immunized (wk 0 and 4) with 10 μg HSA or HSA-PenG in alum. Created with BioRender.com released under a Creative Commons Attribution-NonCommercial-NoDerivs 4.0 International license. **e** Terminal HEL-specific IgG EPT was evaluated. PenG-specific endpoint titres were evaluated

by screening cross-reactivity against HEL-PenG. IgG titres against PenG were evaluated both (**f**) longitudinally and (**g**) at the terminal timepoint. **h** Competition ELISA was conducted, wherein HSA-PenG antisera binding for HEL-PenG was competed out with soluble PenG. **i** Longitudinal protein backbone-specific, HSA, and (**j**) terminal HSA-PenG-specific IgG endpoint titres were screened. **e–j** Dots represent data from a single animal ($n = 4$ per group), and bars/text denotes the median (ND = not detected). Groups were compared via Mann–Whitney test (two-sided). Source data are provided as a Source Data file.

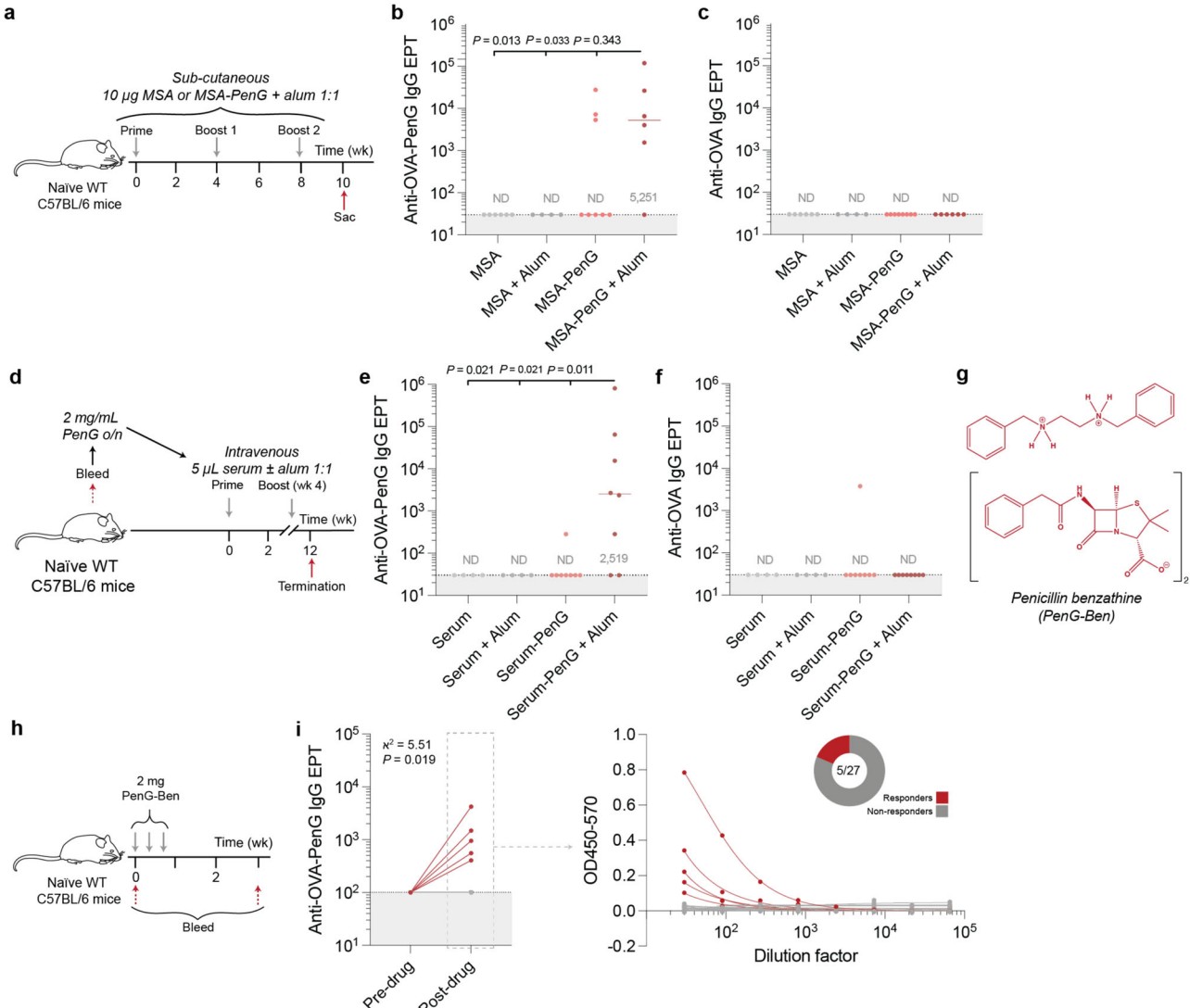

**Fig. 2 | Factors affecting the PenG-specific serological response. a** Sex-matched C57BL/6 mice were immunized three times and the serological response at the terminal timepoint (wk 10) was evaluated. **b, c** Mice were immunized three times (wk 0, 4 and 8) and the terminal (wk 10) IgG PenG-specific EPTs were evaluated. **d** Mice were bled and 2 mg/mL PenG was added and mixed end-to-end overnight. Animals were subsequently immunized (wk 0 and 4) intravenously with seum-PenG with or without alum. **e, f** Terminal IgG EPTs were evaluated. **b, c, e, f** Dots represent data from a single animal ($n = 4$–8 per group), and bars/text denotes the median

(ND = not detected). Data were compared via a Kruskal–Wallis test (two-sided). **g** PenG-Ben structure. **h** Mice were given PenG-Ben intramuscularly. **i** PenG-specific IgG titres were evaluated. Dots represent data from a single animal ($n = 27$). Data were compared to pre-administration, evaluating the ratio of responders via Chi-squared test. **a, d, h** Created with BioRender.com released under a Creative Commons Attribution-NonCommercial-NoDerivs 4.0 International license. Source data are provided as a Source Data file.

## Immunogenicity of free penicillin via varying administration routes

Having shown that PenG is immunogenic when conjugated to diverse protein carriers including mouse serum, we next evaluated whether free penicillin, as would be administered in the clinic, might be sufficient to induce an antigen-specific antibody response. First, we tested whether PenG delivered i.v. daily to mice in $2 \times 1$ week-long courses was immunogenic (Fig S5a). However, no IgG or IgM drug-specific responses were detected when compared to mice given control PBS (Fig S5b, c). Second, the immunogenicity of orally administered antibiotic was evaluated, using the gut-stable oxo-homologue penicillin V (PenV). Unlike PenG, PenV is used for oral administration as it does not degrade under the acidic conditions of the stomach[36]. Mice were given

use, is likely required under such conditions to overcome a threshold for immunogenicity.

PenV *ad libitum* for two 3.5-day intervals. Some mice were additionally given an i.p. dose of lipopolysaccharide (LPS) (0.5 mg/kg) to mimic possible systemic increase in endotoxin expected from a bacterial infection, where antibiotics such as PenG/V would be clinically used (Fig S5d). Despite this, no specific PenG titres were observed (Fig S5e). Notably, administration of LPS increased the background reactivity of serum: mice given PenG and LPS or drug-free water and LPS-only both exhibited modest reactivity against HSA-PenG, which we attribute to induction of B cells producing polyspecific IgG responses[37].

Free drugs can be rapidly cleared; for instance, mice have an extremely high cardiac output[38]—renal clearance of PenG is efficient[39], with a previously reported half-life of approximately 15 min. Fast clearance will restrict in vivo PenG adduct formation, ultimately reducing the probability for antigen-B cell encounter and BCR cross-linking. Therefore, to extend the availability of free drug in vivo, penicillin G benzathine (PenG-Ben) was used. This is a formulation of

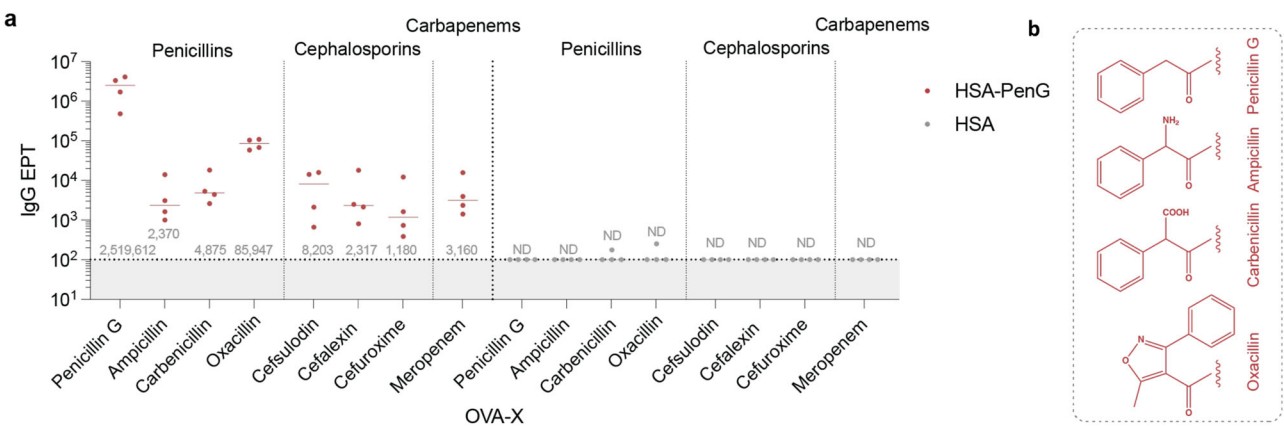

**Fig. 3 | Specificity and cross-reactivity in the PenG-specific antibody response. a** HSA-PenG antisera were screened against a set of β-lactam antibiotic-modified OVA. Data reflect the IgG EPT against the drug adducts, with dots denoting reactivity from a single animal ($n = 4$), with bars/text denoting the median. **b** Sidechains of penicillins tested. Source data are provided as a Source Data file.

PenG as its dibenzylethylene diammonium salt that renders PenG effective for use in slow-release delivery (Fig. 2g). When administered intramuscularly, PenG-Ben is solubilised over days to weeks, to release PenG systemically[40]. PenG-Ben was administered intramuscularly (i.m.) to 27 mice (Fig. 2h). After the antibiotic course, 5/27 (18%) mice generated IgG responses to OVA-PenG, significantly higher than the proportion from pre-immune serum ($\chi^2 = 5.51$, $P = 0.019$) (Fig. 2i). Reactivity against unmodified OVA was not detected (Fig S5f).

**Cross-reactivity of anti-penicillin-adduct IgG responses are drug side-chain and core focused**

PenG, of course, shares common structural homology with the other penicillins as well as other β-lactam antibiotics. We therefore screened the cross-reactivity of the HSA-PenG serological response. Ovalbumin (OVA) was modified with a set of penicillin antibiotics with differing side chains (Fig. 3), and with β-lactam antibiotics from other classes (including cephalosporins and carbapenems), using the previously determined conditions (Fig. 1c). Extent of modification by drug was confirmed by evaluating reduced primary amine availability (Fig S6). Autologous reactivity against OVA-PenG was the greatest of the diverse OVA-X panel tested, with a median IgG EPT of ~$2.5 \times 10^6$ (Fig. 3a). Interestingly, there was limited reactivity against OVA-ampicillin (median EPT of ~$2.4 \times 10^3$), which differs only in a single benzylic amine substituent, and similarly carbenicillin (median EPT of ~$4.9 \times 10^3$), which differs by its benzylic carboxyl substituent. However, there was considerable cross-reactivity against OVA-oxacillin (median EPT of ~$8.5 \times 10^4$), despite the greater variation in sidechain compared with ampicillin. These data suggest that the polyclonal response tolerates some change in side-chain but that this may also be blocked by simple alterations at pivotal sites (such as the ampicillin H→NH$_2$, or the carbenicillin H→COOH change). A subset of 1st–4th generation cephalosporin- and carbapenem-type antibiotics were also screened for cross-reactivity. HSA-PenG antisera displayed limited (albeit above the detection limit) cross-reactivity against these modified OVA antigens (IgG EPTs ~$10^3$). Since cephalosporins and carbapenems have differing β-lactam-encompassing cores[41], these data suggested that the PenG-raised antibody response may be in part dependent on the 6-aminopenicillanic acid-derived core.

**Clonotypic B cell responses to PenG adducts**

To evaluate the B cell response at the clonal and molecular levels, PenG-specific B cells were isolated and variable regions cloned using techniques previously described[42,43]. Mice were immunised with HSA-PenG in alum and draining inguinal lymph nodes (iLN) were harvested 2 weeks post-prime (Fig. 4a). To isolate PenG-specific B cells, requisite

protein-based probes were synthesised by modifying another carrier protein (HIV-1 gp120) that we have validated as giving low background and high specificity in a other hapten-carrier contexts[29]. Gp120 was modified with PenG and then modified with fluorophore Alexa Fluor 647 using a corresponding NHS ester. We further tetramerised biotinylated gp120 with streptavidin-phycoerythrin. These antigen-displaying probes were then used to sort the PenG-specific B cells on pre-gated non-naïve (DUMP$^-$B220$^+$IgD$^-$) B cells (Fig. 4b; Fig S7a). B cells were sorted from four mice and cell clonality was inferred according to the VH sequences (Fig. 4c).

Considerable sharing of V$_H$ gene segments was observed between the mice, suggesting that similar clonotypes were raised across animals (Fig. 4d; Fig S7b). Notably, the V$_H$ gene segments utilised were from four highly phylogenetically related subgroups: *IGHV2*, *IGHV5*, *IGHV10* and *IGHV14*. This striking homology is reflected in conservation of the CDRH1 and CDRH2 amino acid sequences across the mice (Fig. 4e). Together, these data suggest that there are preferred structural and functional motifs encoded in these germline segments that facilitate binding with the drug. Immunogenetic analysis revealed a defined and ordered CDRH3; the length was bimodal, either short (5–6 aa) or long (17 aa) (Fig S8a). Short CDRH3s were dominated by an ARG motif for the first three residues, with a diverse C-terminal end, while the long class possesses a negative N-terminus, a neutral centre and a positive C-terminus (Fig S8b).

Clonal families were evaluated. The two largest families were isolated from mouse 1 and their germinal centre trees determined (Fig S9). The largest clonal family (Fig S6a) exhibits a 'clonal burst' following the acquisition of the S11T mutation, a characteristic phenomenon reportedly associated with the acquisition of an affinity-improving mutation that renders the clone more competitive for antigen uptake and T cell help[44]. This mutation was also found in a separate clade on the same tree. The T85S mutation was also identified twice on separate clades.

Finally, to validate the PenG specificity of the antibody response, a subset of V-region pairs from all mice and with diverse gene segment origins were cloned, and corresponding fragment antigen-binding regions (FAbs) were expressed and purified. Binding was validated via ELISA; all FAbs bound the PenG adduct probe, while an unrelated antibody FAb (BAR-1) did not detectably bind (Fig. 4f–h). These data confirmed that the sorting approach was highly specific.

**Structural, biochemical and biophysical characterisation of the antibody response to PenG**

We selected a subset of PenG-specific clones with divergent, representative CDRH3 lengths to further dissect the binding characteristics

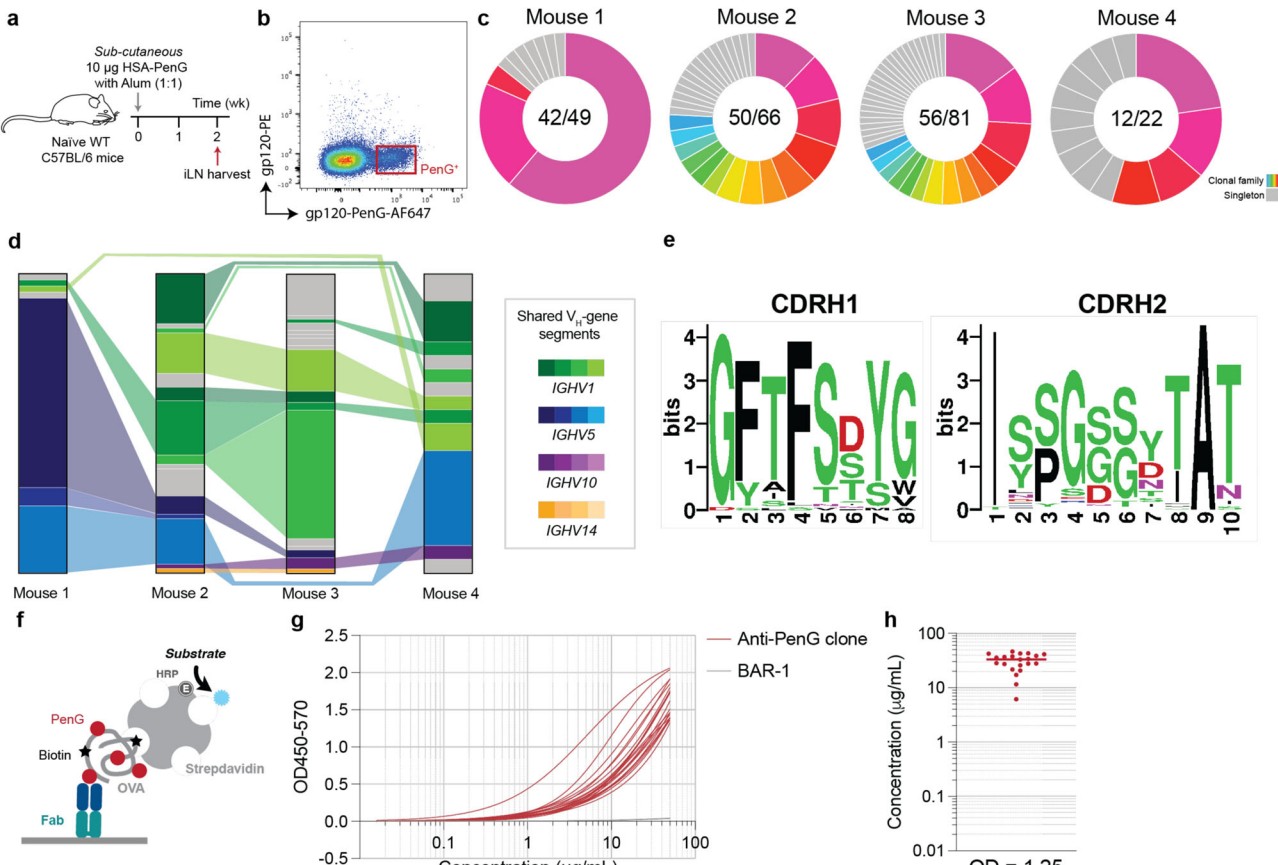

**Fig. 4 | Immunogenetic characterisation of the PenG-specific response.**
**a** Immunisation schedule. Created with BioRender.com released under a Creative Commons Attribution-NonCommercial-NoDerivs 4.0 International license.
**b** Antigen probe sorting strategy on pre-gated non-naïve B cells (DUMP⁻B220+IgD⁻). **c** Inferred clonal families from the PenG probe-sorted B cells (coloured) and singletons (grey). **d** Inferred V_H gene segment across the sequenced B cells. Bar segment sizes proportional to the number of B cells of the same $V_H$ origin. Joining of the segments connote the shared utilisation between mice. Segment colours reflect the *IGHV* subgroup. **e** Logo plots of the CDRH1 and CDRH2 amino acid sequences from all sequenced B cells in all animals. **f** Fab binding specificity assay setup. **g** ELISA trace for a subset of 28 Fabs and (**h**) their OD1.25 intersection (estimate for $EC_{50}$) values. BAR-1 (grey) is an unrelated negative control specific to sialyllactose. Dots represent data from a single Fab clone (*n* = 28). Source data are provided as a Source Data file.

of the clonotypical response to penicilloyl adducts: MIL-1 (*IGHV5-6*01*; 17 a.a. CDRH3), MIL-2 (*IGHV5-17*01*; 6 a.a. CDRH3) and MIL-3 (*IGHV10-3*01*; 9 a.a. CDRH3). We designed a soluble Lys-PenG ligand as a reductionist adduct mimic reflecting β-lactam display on the ε-amine of lysine residues (Fig. 5a).

First, atomic resolution of the binding pose was dissected by protein NMR using universal saturation transfer analysis (uSTA)[29,45]. High transfer efficiencies were observed on the phenyl ring of Lys-PenG with all three FAb fragments, with the *p*-proton showing the greatest engagement for FAbs MIL-1 and MIL-3 (Fig. 5b,c), suggestive of an end-on binding mode for the PenG phenylacetamide sidechain. Notably, the Lys residue itself exhibited minimal transfer efficiency in all cases, confirming, as was implied by the immunological data, that binding is dominated by drug adduct rather than peptide binding. Interestingly, the thiazolidine ring showed no significant engagement of the protons that are detected by uSTA. This engagement was further confirmed by drug adduct core binding (although less than drug sidechain) through observed transfer efficiencies of the proton at the stereogenic centre of the opened *β*-lactam ring (NC*H*) as well as those of the benzylic sidechain $CH_2$. These data revealed substantial uniformity in the binding poses adopted by the MIL series of antibodies.

Next, we determined the X-ray crystal structure of the PenG-Lys•MIL-3 complex at 2.2 Å (Table S1). Three FAb molecules were present in the asymmetric unit (H (heavy)/L (light), A/B and C/D). In both H/L and A/B molecules, the phenylacetamide PenG sidechain and the thiazolidine moieties are well-defined in the electron density whilst the Cβ, Cδ and Cγ portion of the lysine moiety has weak density, indicating it is less well ordered and consistent with our observations by protein NMR (see above). The electron density is considerably weaker in A/B than in H/L most likely due to crystal contacts present in H/L (Fig S10a, Fig S11). Apart from this region, the interactions between the ligand and the protein are conserved in both FAbs. In the third FAb molecule (C/D), CDRH loops are highly disordered, and no ligand was fitted.

Our analysis focuses on the H/L molecules (Fig. 5f). The PenG phenyl group is deeply buried in a narrow hydrophobic pocket sandwiched on one face of the benzene ring by CDR L loops (Tyr68_L, Tyr110_L) and the β-turn formed by CRDH3, mainly the side chain of Ile120_H with some involvement of the main chain of Phe124_H, on the other face (Fig. 5f ii., FigS10b). The interaction with Tyr68_L has strong π-stacking character whilst the planes of the rings found in the phenyl moiety and Tyr110_L are offset 70° and thus hydrophobic in character. The methyl group of Cγ2 Ile120_H sits centred above plane of the ring.

The static X-ray structure does not disclose a simple explanation for the observation from NMR for the interactions of the *p*-hydrogen yet provides detail on PenG sidechain recognition. The nitrogen of the phenylacetamide forms a hydrogen bond with the carbonyl of Thr121_H whilst the phenylacetamide carbonyl oxygen bridges via a water molecule (W1) to the side chain hydroxyl of Tyr110_L. The β-turn conformation of CDRH3 is stabilised by hydrogen bonds between residues

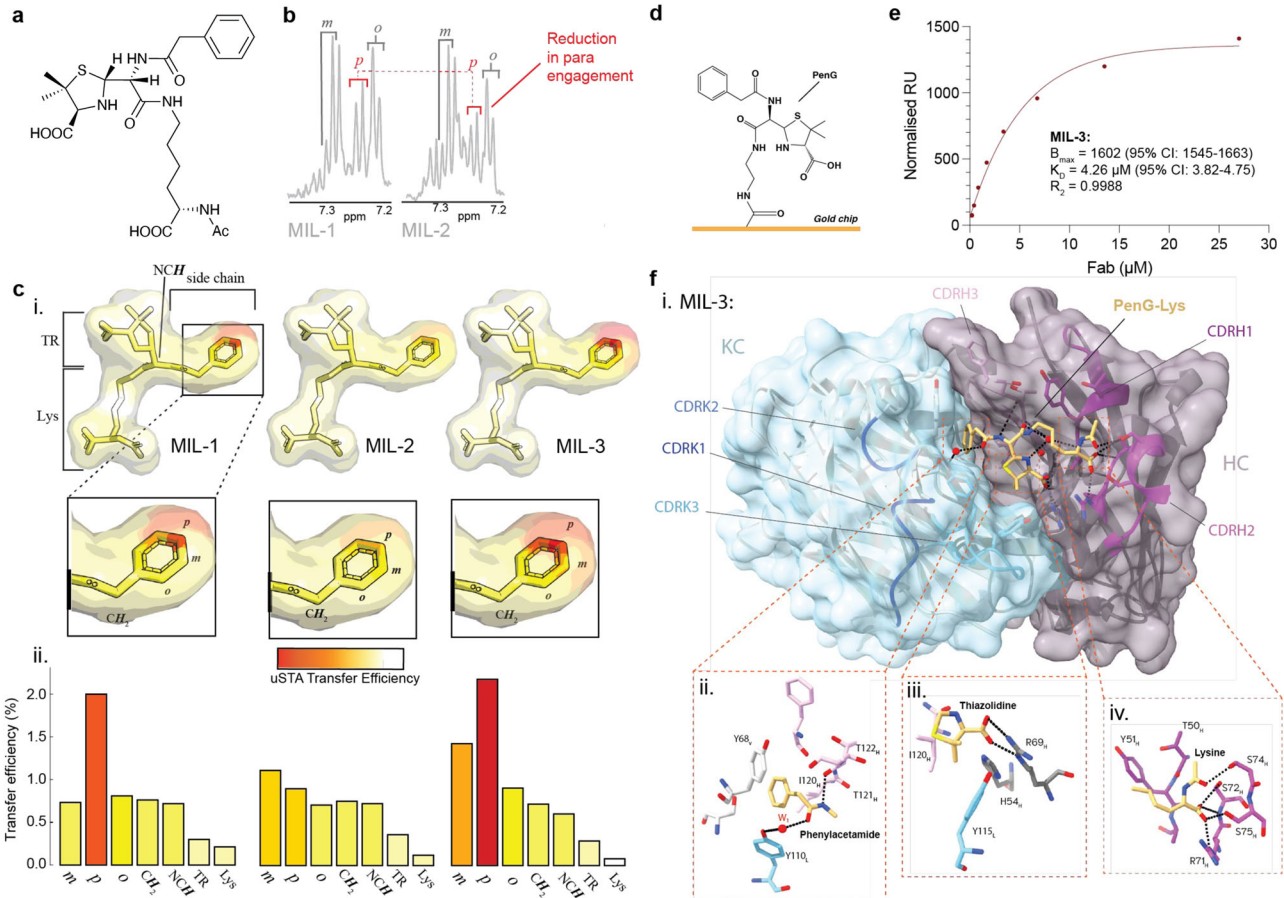

**Fig. 5 | Structural, biochemical and biophysical characterisation of the PenG-specific response. a** Annotated chemical structure of PenG-lys construct used for uSTA analysis. **b** Saturation transfer difference spectrum was generated from the difference between the raw off-resonance (gaussian at 37 ppm) and the raw on-resonance (gaussian at 9 ppm) spectra. Data showing differences in para engagement in the benzene ring of MIL-2 fab versus MIL-1. **c** i. Heatmaps corresponding to saturation transfer efficiency of PenG-lys (1 mM) with MIL-1–3 (5 μM). ii.

Histographic saturation transfer efficiencies. Red indicates high transfer efficiency. **d** SPR chip design. **e** Biophysical characterization of MIL-3 via SPR. **f** i. Top view of the x-ray structure of the MIL-3 Fab bound to the PenG-Lys (beige sticks). Both heavy (monochrome plum) and light chain (monochrome blue) CDRs are marked. Key residues within 4.0 Å of ligand aspects ii. phenylacetamide, iii. thiazolidine and iv. lysine are shown. Hydrogen bonds are shown as black broken lines. Water is marked in red. Source data are provided as a Source Data file.

Thr122$_H$ and Arg123$_H$ and Tyr 51$_H$ and Asp75$_L$. The carboxylate of the thiazolidine moiety makes a bidentate salt bridge to the guanidine group of Arg69$_H$ and potentially a salt bridge with His54$_H$ (Fig. 5f iii.); this binding mode is likely to powerfully contribute to binding enthalpy yet places the protons that are observable by NMR more remotely. The dimethyl group and thiazolidine ring make van der Waal contacts with Tyr115$_L$ and Ile120$_H$. The nitrogen of the thiazolidine ring interacts with a highly coordinated water molecule (W3) bridging CDRs H1, H3 and the lysine linker. The main-chain mimic of the Lys makes five hydrogen bonds with the 3$_{10}$ helix of CDRH2 in H/L (Fig. 5f iv.) but only one in A/B (Fig S10c), consistent with differences in ordering and previously noted weaker interactions (see above). Torsion angle modification of the lysine side chain would permit conjugated protein to remain outside the binding pocket without any perturbation of the drug adduct thiazolidine and phenylacetamide interactions.

Finally, surface plasmon resonance (SPR) of FAb MIL-3 with a PenG-adduct chip revealed a $K_D$ value = 5.3 μM (90% CI: 3.996–7.37, Fig. 5d,e).

## Antibody clones are unlikely to sequester antibiotic during relevant treatments

The biophysical properties of the antibodies that we determined, the structural biology and the physiological concentrations of antibody

in vivo all suggested that anti-PenG-adduct antibodies are unlikely to display any drug-sequestering effects. Nevertheless, this was evaluated by developing a model system to test whether PenG-specific antibodies could reduce the bacterial growth-inhibiting function of PenG. Whole antiserum was added to a culture of attenuated, unencapsulated *Streptococcus pneumoniae*[46], chosen because of its high sensitivity to PenG (minimum inhibitory concentration (MIC) = 0.01 μg/mL). We conducted two assays: first, we evaluated the effects of anti-HSA-PenG antiserum on PenG sequestration in culture and, second, contrasted PenG kill zones when drug was pre-incubated with high quantities of purified recombinant control or anti-PenG MIL antibodies (Fig S12). As expected, neither antiserum nor recombinant MIL antibody significantly inhibited antibiotic action.

## Discussion

The elicitation of ADA such as those against PenG may have broad implications, including the mediation of drug hypersensitivity[11,18,47,48]. The complex relationship between chemical reactivity and instability, pharmacokinetics, immunological factors and possible downstream functional effects of the resulting humoral responses are poorly understood. To address this, we have systematically evaluated the way in which the protein conjugation reactivity of the common β-lactam antibiotic, PenG, drives formation of antigenic complexes in vitro, and have established that diverse protein carriers are sufficient to

propagate a drug-specific IgG response. Using a murine model, we characterised both the clonal B cell and antibody responses revealing striking clonal restriction and strong conservation of antibody-drug binding features despite diversity in CDRH3 length. Our data demonstrate that the production of PenG-specific antibodies is regulated at two distinct levels: (1) the covalent formation of protein adducts via lysine-amide formation is influenced by reaction conditions and time, and (2) immune engagement, including innate recruitment−such as via adjuvant which in vivo would be mimicked by bacterially-elicited inflammation−and T cell help. These factors ultimately determine the overall probability and magnitude of a downstream B cell response against PenG. Collectively, our data now provide a model for how adduct formation and immune engagement trigger the humoral response, phenomena that may inform both the study of allergy and potentially provide a rational approach to predict and even mitigate anti-penicillin antibody responses.

In a human context, most patients undergoing standard courses of PenG to treat bacterial infection exhibit anti-penicilloyl IgG antibodies thereafter[10,49,50]. These data appear initially incongruent with our murine data since animals given formulations of free penicillin either intravenously or in drinking water failed to exhibit a specific response. However, it is noteworthy that these selective effects may be attributed to dosing; humans are given as much as 50 mg/kg every 4−6 h of PenG via an intravenous line[51,52]. Moreover, higher murine cardiac output and drug clearance rate results in shorter drug half-life[38,39]. A critical role for PenG concentration, which will be higher in human circulation, in driving relevant adduct formation is supported by the serological data gathered from mice given both static preformed serum-PenG immunisations and intramuscular slow-release PenG-Ben. We did not titrate the downstream differences in murine versus human adaptive immune responses to penicillin, although differences are reported in the literature, particularly T cell-mediated responses to drug adducts[14].

These experiments were not designed to explicitly evaluate allergic outcomes in mice. Isotype-switching to IgE and the downstream mast cell-mediated and other modes of allergic reactivity are determined by both genetic factors in the form of atopic predisposition to IgE production, and environmental factors[53,54]. However, the functional effect of the antibody response can not only be a feature of the Fc effector function or isotype−for example, whether type I versus type II hypersensitivity is imparted−but also the inherent binding mode and biophysical properties of the antibody. Our data now show that the murine B cell repertoire responds to the drug adduct with a dominant clonotypic family that primarily engages the side-chain constituent, phenylacetamide, and the carboxylate group of the thiazolidine, as evidenced by our complementary uSTA[45] and x-ray structural characterisation. These data point to a striking homogeneity of response, implying that B cell receptors and subsequent antibodies may be restricted in the binding solutions that they can adopt to recognise such a small non-protein antigen. This is consistent with previous immunological mapping studies of murine[55,56], rabbit[27] and human[57] antibody responses in which side-chain reactivity appears prominent. Importantly, the biophysical features of these antibodies, with monovalent FAb $K_D$ in the low−mid µM range and a relatively fast $k_{off}$, coupled with relatively low concentrations in vivo, are likely to discount even partial drug inhibitory effects.

Our approach, applied here to an archetypal inhibitor with a covalent bond-forming mode-of-action, now creates a potential roadmap for understanding the chemical, pharmacological and immunological factors governing whether a B cell response can be mounted against other protein-reactive covalent therapeutics, for example, the amide-forming acylating drug aspirin[58]. Having identified a dominant clonotype against the PenG adduct, this excitingly suggests a workflow that could be used as a model to 'reverse engineer' a PenG analogue that fails to engage the MIL clonotype, potentially reflecting a low/no

allergenic alternative against a murine germline and that now instructs a proof of principle for germline-informed drug design in humans.

## Methods

### Ethics and permissions
All experiments were conducted under approved licenses and protocols, consistent with national (as authorised by the Home Office of the United Kingdom) regulations and University of Oxford guidelines.

### Ex vivo modification of protein with β-lactam antibiotics
Carrier proteins (HEL, OVA, HSA, BSA and MSA) were purchased commercially (Merck) and dissolved in 0.1 M NaCO$_3$ (pH = 8, unless otherwise indicated) and concentration was adjusted to 1 mg/mL. β-lactam was added to a molar ratio of 1:200 per lysine residue of carrier protein, resulting in protein concentrations for HEL and HSA of 6.8 and 60 mg/mL respectively. The mixture was rotated end-to-end at 25 °C overnight and dialysed into PBS.

### Denaturing MS
Reversed-phase chromatography was performed in-line prior to mass spectrometry using an Agilent 1100 HPLC system (Agilent Technologies inc.−Palo Alto, CA, USA). Concentrated protein samples were diluted to 0.02 mg/ml in 0.1% formic acid and 50 µl was injected on to a 2.1 mm ×12.5 mm Zorbax 5um 300SB-C3 guard column housed in a column oven set at 40 °C. The solvent system used consisted of 0.1% formic acid in ultra-high purity water (Millipore) (solvent A) and 0.1% formic acid in methanol (LC-MS grade, Chromasolve) (solvent B). Chromatography was performed as follows: Initial conditions were 90% A and 10% B and a flow rate of 1.0 mL/min. After 15 s at 10% B, a two-stage linear gradient from 10% B to 80% B was applied, over 45 s and then from 80% B to 95% B over 3 s. Elution then proceeded isocratically at 95% B for 1 mins 12 s followed by equilibration at initial conditions for a further 45 s. Protein intact mass was determined using a 1969 MSD-ToF electrospray ionisation orthogonal time-of-flight mass spectrometer (Agilent Technologies Inc.−Palo Alto, CA, USA). The instrument was configured with the standard ESI source and operated in positive ion mode. The ion source was operated with the capillary voltage at 4000 V, nebulizer pressure at 60 psig, drying gas at 350 °C and drying gas flow rate at 12 L/min. The instrument ion optic voltages were as follows: fragmentor 250 V, skimmer 60 V and octopole RF 250 V. Obtained MS spectra were processed and deconvoluted using the Agilent MassHunter Qualitative Analysis (B.07.00) software.

### LC-MS/MS
Approximately 5 µg protein was reduced, loaded and run on an SDS-PAGE. Gel bands were excised and washed sequentially with HPLC grade water followed by 1:1 (v/v) MeCN/H$_2$O. Gel bands were dried (via vacuum centrifuge), treated with 10 mM dithiothreitol (DTT) in 100 mM NH$_4$HCO$_3$ and incubated for 45 min at 56 °C with shaking. DTT was removed and 55 mM iodoacetamide (in 100 mM NH4HCO3) was added and incubated for 30 min in the dark. All liquid was removed and gels were washed with 100 mM NH$_4$HCO$_3$/MeCN as above. Gels were dried and 12.5 ng/µL trypsin was added separately and incubated overnight at 37 °C. Samples were then washed and peptides were extracted and pooled with sequential washes with 5% (v/v) formic acid (FA) in H$_2$O and MeCN. Dried samples were reconstituted in 2% MeCN 0.05% trifluoroacetic acid and run by LC-MS.

Samples were analysed using an Ultimate 3000 UHPLC coupled to an Orbitrap Q Exactive mass spectrometer (Thermo Fisher Scientific). Peptides were loaded onto a 75 µm × 2 cm pre-column and separated on a 75 µm × 15 cm Pepmap C18 analytical column (Thermo Fisher Scientific). Buffer A was 0.1% FA in H$_2$O and buffer B was 0.1% FA in 80% MeCN with 20% H$_2$O. A 40-min linear gradient (0% to 40% buffer B) was used. A universal HCD identification method was used. Data was collected in data-dependent acquisition mode with a mass range 375 to

1500 m/z and at a resolution of 70,000. For MS/MS scans, stepped HCD normalized energy was set to 27, 30 and 33% with orbitrap detection at a resolution of 35,000.

Raw data was first searched using the FragPipe (v19.1)[59] Open Search pipeline to determine the modified mass shift caused by PenG conjugation. Approximately 2.3% of peptide spectral matches (PSM) had an unannotated mass shift of 334.099 Da. Next, to determine site specificity and occupancy, raw data was searched using Proteome Discoverer (3.0.0.757). In-house curated FASTA databases were used. The digestion enzyme was set to trypsin with a maximum of 2 miss cleavages. A 10 ppm precursor mass tolerance and 0.6 Da fragment mass tolerance were allowed. Oxidation (+15.995 Da) of methionine and PenG conjugation (+334.099 Da) of lysines and protein N-termini were set to dynamic modifications. Carbamidomethylation (+57.021 Da) of cysteines was set as a static modification. Target false discovery rate (FDR) for peptide spectrum matches, peptide and protein identification was set to 1%. To approximate site-specific occupancy of PenG conjugation the total number of PSMs of peptides containing a specific lysine site in a PenG modified state were expressed as a percentage compared to the total number of PSMs containing the given lysine in both the modified and unmodified state.

To estimate lysine reactivity in HSA, raw data was searched against a database of known contaminants which contains the canonical HSA sequence using FragPipe (21.1) using a standard closed search parameters with an additional variable modification confined to lysines (+334.099). IonQuant (as implemented in FragPipe) was used to calculate peptide intensities with default parameters. Intensity of each modification-specific peptide was normalised against total intensity of all HSA peptides in treated and control samples. For each position, a ratio of intensity of unmodified peptide (without missed cleavages, as present in control sample) between the treated sample and the control was calculated to estimate the extent of lysine modification at that position after treatment.

LC-MS/MS raw data and search results have been deposited to the ProteomeXchange Consortium (http://proteomecentral.proteomexchange.org) via the PRIDE partner repository[60] with the dataset identifier: PXD052026.

### Free primary amine ELISA
After the drugs were conjugated to a carrier protein using the method outlined above, the relative abundance of free amines was assessed to determine the extent of lysine modification. Protein samples (5 μg) were dissolved in 10 μL of PBS and mixed with 40 μL of 0.1 M sodium bicarbonate buffer. A 5% solution of 2,4,6-trinitrobenzenesulfonic acid (TNBSA) was diluted at a ratio of 1:500 in the bicarbonate buffer, and 25 μL of this mixture was added to the protein samples. Following a 2 h incubation period at 37 °C, 25 μL of 10% SDS and 12.5 μL of 1 M HCl were added. The absorbance at 335 nm was then measured.

### Mice and immunisation formulations
Wild-type, specific pathogen-free, sex-matched, 6–8-week-old C57BL/6 mice were purchased from Charles River. Animals were monitored daily and were provided standard chow and water *ad libitum*. Immunisation formulations and schedules are outlined in the results. Mice were bled periodically from the tail vein. Animals were sacrificed via a rising $CO_2$ gradient and subsequent cervical dislocation schedule 1 procedure.

### ELISA
Serum samples were serially diluted and transferred onto an antigen-coated and blocked SpectraPlate-96 (PerkinElmer) plate. Binding was detected with an anti-mouse IgG-HRP (STAR120P, Bio-Rad). ELISAs were developed using 1-Step-Ultra TMB ELISA substrate (Life Technologies), terminating the reaction with 0.5 M $H_2SO_4$. For competition ELISAs, serial dilution of soluble ligands as preincubated with

antiserum at the pre-determined $EC_{50}$ concentration for 1 h. The antisera and ligand mixtures were subsequently transferred onto the antigen-coated and blocked plates and ELISA conducted, as previously outlined. Cytokine ELISAs were performed using commercially available kits (Life Technologies), screening supernatant from antigen-restimulated splenocytes.

Optical densities were measured at 450 and 570 nm on a Spectramax M5 plate reader (Molecular Devices). After background subtraction, logistic dose-response curves were fitted in GraphPad Prism. Endpoint titres were determined as the point at which the best-fit curve reached an $OD_{450-570}$ value of 0.01, a value which was always > 2 standard deviations above background.

### B cell sorting
Penicilloyl-specific B cells were isolated using antigen probes. Gp120-PenG was modified with an NHS-esterified AF647 dye, as per the manufacturer's instructions (Life Technologies). To improve true antigen-specific cell sorting efficiency, a negative backbone-specific probe was synthesised, wherein biotinylated gp120 was tetramerised with PE-conjugated strepdavidin (Biolegend).

Single cell suspensions were stained with LIVE/DEAD Fixable Blue and Fc receptors blocked. Surface staining was performed using anti-mouse F4/80-PE (1:200, BM8, Biolegend), anti-mouse Gr-1 (1:200, RB6-8C5, Biolegend), anti-mouse CD3-PE (1:200, 17A2, Biolegend), anti-mouse CD4-PE (1:200, RM4-5, Biolegend), anti-mouse CD8-PE (1:200, RPA-T8, Biolegend), anti-mouse B220-eFluor450 (1:100, RA3-6B2, BD Biosciences), anti-mouse IgD-AF700 (1:200, 11-26 c.2a, Biolegend), anti-mouse IgM-PE/Cy7 (1:200, R6-60.2, BD Biosciences), anti-mouse IgG1-FITC (1:200, A85-1, BD Biosciences), anti-mouse IgG2a/2b-FITC (1:200, R2-40, BD Bioscience) and antigen probes (10 μg/mL). Cells were stained on ice for 1 h, washed and stored on a BD FACSAriaFusion (BD Biosciences). Single cells were sorted into MicroAmp Optical 96-well PCR plates (Life Technologies), isolating LIVE/DEAD⁻DUMP⁻B220⁺IgD⁻gp120⁻gp120-PenG⁺ events. Cells were sorted directly into 5 μL if 1X TCK buffer supplemented with 1% 2-ME and stored at −80 °C until use.

### Thymidine incorporation
Whole splenocytes were stimulated in vitro with 10 μg/mL antigen in cRMPI for 16 h in a flat-bottom 96-well plate. During the final 6 h stimulation, each well was spiked with 0.037 mBq tritiated thymidine (Perkin Elmer). Cells were transferred and lysed on glass filter mats (Perkin Elmer) using a Micro 96 Harvester (Skatron Instruments). Tritium incorporation was measured using Betaplate Scint and a Microbeta Trilux Scintilation counter (Perkin Elmer).

### Intracellular cytokine staining
Whole splenocytes were stimulated in vitro with 10 μg/mL antigen in cRPMI for 16 h. For the final 6 h, 5 μg/mL brefeldin A (Biolegend) was added to suspend ET−Golgi trafficking and block cytokine secretion. Cells were stained with TruStain mouse FcX Plus (Biolegend) and LIVE/DEAD Fixable Blue in PBS with 2 mM EDTA for 30 mins. Surface markers were subsequently stained: PE-conjugated anti-mouse CD3 (dilution: 1:200, clone: 17A2, manufacturer: Biolegend), APC-conjugated anti-mouse CD4 (1:200, RM4-5, Biolegend), AF700-conjugated anti-mouse CD8 (1:200, RPA-T8, Biolegend). Following fixation and permeabilization (Biolegend), cells were stained with PE/DAZZLE-conjugated anti-mouse IFN-γ (1:100, XMG1.2, Biolegend). Cells were washed and data was acquired on the BD Fortessa X-20 (BD Biosciences), collecting 500,000 events per sample.

### Variable region cloning and antibody expression
B cell receptor variable regions were recovered, as previously described[29]. Briefly, RNA was captured on RNAClean XP beads (Beckman Coulter) and washed with 70% ethanol. RNA was eluted and cDNA

was synthesised using SuperScript III (Life Technologies) with random primers (Life Technologies). VH and VK regions were recovered[43] and Q5 polymerase (New England Bioscience), sequencing the amplicons via Sanger. VH amplicon sequences were used to determine B cell clonality.

To validate that the sequences were specific to the penicilloyl adducts, antibodies were recombinantly expressed. The VH/VK amplicons were incorporated into expression vectors: vector-overlapping adapters were incorporated via PCR[42], and the V regions were inserted into pre-cut recombinant FAb expression[44] vector via Gibson reaction (New England Bioscience). Vector products were transiently transfected into HEK 293Freestyle cells and FAb was purified from supernatant using Ni-NTA resin.

## Immunogenetic analyses

Analyses were performed using VH regions. Sequences were aligned to the murine reference genome using the Immunogenetics Information System (IMGT; https://www.imgt.org/IMGT_vquest/input), as described previously[29]. Sequence outputs of poor quality or those unproductive were excluded from our analyses. Alignments of CDRs were visualised using WebLogo[61]. Clonal lineages were evaluated using GCTree[62].

## SPR

SPR was performed using a Biacore T200 instrument. Details of chip design, synthesis and testing, refer to Document S1. FAb binding was evaluated by sequentially injecting serial dilutions at a flow rate of 10 μL/min.

## uSTA

Samples for uSTA were prepared by buffer-exchange of purified FAb fragments with $D_2O$ PBS using Amicon 30 K MWCO. All NMR experiments were conducted on Bruker Avance Neo 600 MHz spectrometer at 25 °C equipped with QCIF cryoprobe and a SampleJet, running TopSpin 4.2.0. The uSTA experiments were recorded using a pseudo-3D pulse sequence based on stddiffesgp.2 from the standard Bruker library as described previously[45]. The following saturation times were used: 0.1, 0.3, 0.5, 0.7, 0.9, 1.1, 1.3, 1.5, 1.7, 1.9, 2, 2.5, 3, 3.5, 4 and 5 s. Low-power gaussian excitation pulse was applied at 9 ppm and 37 ppm for the on- and off-resonance spectra respectively, where the specific choice of excitation at 9 ppm minimised artefacts in a ligand-only spectrum[45] (Fig. S1). All experiments were recorded with 16 scans per transient, 32768 complex points and sweep width of 16.01 ppm for a total acquisition time of 7 h 41 min. These were acquired on a protein-only, a ligand-only and a mixed protein/ligand sample. Data were processed using the nmrPipe module embedded within the uSTA workflow, where the protein-only data is subtracted from the mixture to give a uSTA transfer efficiently, from which values from the 'ligand only' sample are then subtracted to account for residual ligand excitation, as previously reported[45]. The 'heatmaps' were generated by mapping the uSTA transfer efficiency from the detectable protons onto heteroatoms (carbon, sulphur, oxygen and nitrogen) using a $1/r^6$ dependency, before rendering in pymol as described previously[45].

## X-ray crystallography

MIL-3 FAb was loaded onto a gel filtration Superdex 200 column 10/30 (GE Healthcare) in 10 mM Tris-HCl, pH 7.5, 150 mM NaCl. Co-crystals appeared at 20 C after a week from a hanging drop of 0.1 μL of protein solution (15 mg/mL with 2.5 mM PenG-Lys with 0.1 μL of reservoir solution containing 20% (w/v) PEG 6000, 0.1 M MES pH 6, 0.2 M ammonium chloride in vapor diffusion with reservoir. Crystals were frozen with the same solution containing 16% glycerol. Data were collected at the Diamond light source oxfordshire (beamlines I04). Data were processed with XIA2[63-66]. Structure has been solved by molecular replacement using PHASER and pdb file 7bh8 for VL, CH and CL domains and 7n09 for VH domain. The structure was builded with Autobuild program, refined with REFINE of PHENIX with NCS restraints[67] and adjusted with COOT[68]. Coordinates and topologies of ligands were generated by AceDRG[69].

## Microbiological assays

The attenuated, unencapsulated lab strain *Streptococcus pneumoniae* D39 (Δ*cps2A'*-Δ*cps2H'*)[46] was routinely grown in tryptic soy broth (TSB) (BD Biosciences) at 37 °C (standing incubation) in a 5% $CO_2$ atmosphere.

Microbroth dilutions of S. *pneumoniae* D39 revealed a PenG MIC of 0.01 μg/mL (that is, the lowest antibiotic concentration that prevented bacterial growth) (data not shown). For the liquid antibiotic rescue assay, 2 ng of PenG in 10 μL PBS were pre-incubated with 10 μL of antisera for 2 h in a flat-bottom 96-well plate. 180 μL of exponentially growing S. *pneumoniae* cells ($OD_{600} = 0.2$) were added and incubated overnight (such that the final PenG concentration was 0.01 μg/mL). The following day, bacterial growth was measured using a Spectramax M5 plate reader (Molecular Devices), evaluating the $OD_{600}$ nm as a proxy for bacterial density.

For disk diffusion assays, 0.1 μg of PenG and mAb (1:50 molar ratio) were spotted onto paper disks. The disk was placed onto a blood agar plate (Merck), carrying a 5 ml nutrient soft agar overlay with 200 μL exponentially growing S. *pneumoniae* cells ($OD_{600} = 0.2$). Plates were incubated overnight. The following day, the kill zone diameter was manually measured. PenG-only, PenG-raised antibody clones were tested, as well as an irrelevant HIV-1-specific antibodies were tested.

## Data and statistics

Flow cytometry data was evaluated using FlowJo V.10.8.2 for Mac. Statistical analyses were conducted in either GraphPad Prism V.10.0.1 or in RStudio V.4.1. Statistical test details are provided in the results, figures and associated figure legends.

## Data availability

Data reported in the manuscript are supplied as separate source data files or deposited as otherwise referred throughout or can alternatively be requested directly from the author. LC-MS/MS raw data and search results have been deposited to the ProteomeXchange Consortium (http://proteomecentral.proteomexchange.org) via the PRIDE partner repository[60] with the dataset identifier: PXD052026. The structure of BAR-1 bound to PenG-Lys is deposited in the protein database PDB (https://www.rcsb.org), under the accession number 8QXC. No custom code was developed for this manuscript. Reagents are available where applicable through an institutional MTA agreement. Source data are provided with this paper.

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

## Acknowledgements

The authors thank The Sir William Dunn School of Pathology flow cytometry facility, SPR facility and animal house staff. We extend our gratitude to the Rosetrees Trust, who have supported this work through the Interdisciplinary Award (ID2020/100023). We are also grateful for the funding provided by the Wellcome Trust (224212/Z/21/Z). Additionally, we thank the Wellcome Trust (grant ref: 095872/Z/10/Z) and the Engineering and Physical Sciences Research Council (grant ref: EP/R029849/1) for the instrumental upgrades of the 600-mHz and 950-MHz NMR spectrometers, as well as support from the University of Oxford Institutional Strategic Support Fund, the John Fell Fund, and the Edward Penley Abraham Cephalosporin Fund. A.J.B. is supported by ERC grant (101002859). For the purpose of Open Access, the author has applied a CC BY public copyright license to any Author Accepted Manuscript version arising from this submission. The Chemistry theme at the Rosalind Franklin Institute is sustained by the EPSRC (V011359/1 (P)). We would like to thank the Membrane Protein Laboratory at Diamond Light Source (funded by Wellcome Trust grant 223727/Z/21/Z) for help and support. C.M.K. is supported by an EPA Cephalosporin Junior Research Fellowship from Linacre College Oxford. L.P.D. is supported by the Clarendon Fund, and Q.J.S. is a Jenner Vaccine Institute Investigator and James Martin School Senior Fellow. We are grateful for the technical advice of Anton van der Merwe (The Sir William Dunn School of Pathology, University of Oxford) in the biophysical analyses.

## Author contributions

Conceptualization of project: L.P.D., B.G.D., Q.J.S.; Methodology: L.P.D., L.M., G.S., V.L., A.T., C.J.B., S.A.B., M.K., C.M.K., M.S., W.B.S., A.J.B., J.N., B.G.D., Q.J.S.; Investigation: L.P.D., L.M., G.S., V.L., A.T., C.J.B., S.A.B., M.K., Y.D., C.M.K., W.B.S.; Funding acquisition: M.S., W.B.S., A.J.B., J.N., B.G.D., Q.J.S.; Project administration: M.S., W.B.S., A.J.B., J.N., B.G.D., Q.J.S.; Supervision: S.M., M.S., W.B.S., A.J.B., J.N., B.G.D., Q.J.S.; Writing—original draft: LPD.; Writing—review & editing: L.P.D., B.G.D., Q.J.S.

## Competing interests

The authors declare no competing interests.
