## [Peer Review File · Nature Communications]

REVIEWER COMMENTS

Reviewer #1 (Remarks to the Author):

The manuscript describes a thorough, reductive study of the humoral response to chemical modification of serum proteins by beta-lactam antibiotics in mice. It has been known for many years that small molecules need to be conjugated to carrier proteins to induce a measurable antibody response. It is also known that the reactivity of the beta-lactam ring can lead to the formation of protein-PenG complexes but their immunogenicity is poorly understood. The manuscript therefore addresses an important and under-investigated problem, and its novelty lies in the application of molecular and structural methods to study it. The observation that VH gene segments are highly restricted from deep clonotyping is important, suggesting, as the authors point out, that such information could be used to 'design-out' adverse immunogenic responses in vulnerable individuals. A weakness of the manuscript lies in the fact that this is an idealised mouse model, conducted in uninfected animals. The biochemical and structural studies, which are undoubtedly thorough, are based on the clinically less relevant IgG response. The authors acknowledge that the more serious clinical problems (anaphylaxis) of antibiotic sensitivity are allergic in origin (i.e. IgE). In addition, the binding affinities of the PenG-lys construct used for structural work are comparatively weak- this is exemplified by partial occupancies of the antigen combining sites in the FAb MIL-3 crystal structure, for example. Their relevance to human humoral responses is therefore debateable.

Major

L203: is alum a good adjuvant to use to mimic the effect of bacterial infection? The authors make the point that innate recruitment is likely to be important in the induction of PenG-specific antibodies (L403). Might a Lipid A analog such as MPLA have been better choice of adjuvant?

L243-263: were the reactivities of the various beta-lactam antibiotics to OVA the same? Could the results to some extent be explained by changes in the distribution of reactions with various residues across OVA?

L372: the recorded binding affinity of FAb MIL-3 to a PenG-adduct chip is comparatively weak, at about 5uM. There was no report of the affinities of MIL-1 and MIL-2- it would have been useful to include these data, to provide context for the NMR experiments presented in Fig 5. An argument could be made that a solution binding assay would have been more relevant to the NMR and X-ray crystallography studies (eg using fluorescence) although this is a minor point.

L415: in addition to dosing effects and higher drug clearance rates in mice compared to humans, it also needs to be pointed out that there are differences in immunogenic responses between the two.

Minor

P15 Fig 4 g, h: the line for the BAR-1 control is not very clear in the figure.

L361 'enthalpy yet' (typographical error)

Reviewer #2 (Remarks to the Author):

The authors report an interesting study revealing interaction sites of PenG with protein to elucidate mechanisms of immune responses. They do not only study PenG, but also evaluate different penicillin derivatives as well as other classes such as cephalosporines and the carbapenem meropenem. Moreover, they also double-check the influence in an in vitro system on one *S. pneumoniae* strain. Therefore, they have performed a quite comprehensive study aiming to explain and answer questions of immunogenicity observed in the clinics. Although they explicitly mention, that they did not aim to exactly mimic the clinic, I feel that some aspects might render the manuscript even stronger. I have only 3 major and just some minor comments, which are found below.

Major comments:

(1) For the more clinical setup you chose a dose of 2 mg/ml of PenG for pre-incubation and incubated for 16 hrs. Please specify why you chose that concentration? Is it to mimic peak plasma concentrations? In the clinics constant plasma concentrations would only be achieved during infusion. Have you considered varying the incubation time, also including different (lower) concentrations of PenG to see if the effect will remain similar or if some concentrations might not cause the effect?

(2) Do you have pharmacokinetic information/profiles on PenG, PenV and Benzylpenicillin in your model? You mention a relatively short half-life of PenG. Why did you choose the i.v. route in light of this? The i.m. route gives a different profile. Moreover, there are formulations of PenG which result in longer half-lives after i.v. administration. As your immunogenicity studies are only done with PenG, I would recommend to consider if it was worth to add these data. Moreover, I was wondering which doses for PenG were used and if they were sufficient to achieve the concentrations in plasma you used for your in vitro studies. I am concerned for PenV as you used 2 mg/ml, which does not necessarily translate to 2 mg/ml in vivo. Do you have information if the prodrug PenV is entirely released after intake. Additionally, I am concerned how you make sure that every mouse is getting the same dose to get reproducible results if all animals have access to PenV solution ad libitum. I would appreciate comments on that. In light of that: is there any information available on the PK of Benzylpenicillin after i.m. information with regard to plasma levels? How did you choose the dose of 2 mg?

(3) You provide information on one *S. pneumoniae* strain, which is non-capsulated. Have you tested a capsulated *S. pneumoniae* strain with similar MIC?

Minor comments:

(1) It would be more clear to the reader if you specified in Figure 1b which exact moieties react with which part of the protein or exemplify that. You mention it in the introduction and in the initial sentences of the results, but it would be good to also spot it directly in the Figure.

(2) For the adduct formation in Fig. S1b: please provide MS/MS spectra for the tryptic digest. Moreover, I would encourage to use a schema to show which digestion sites in the protein you actually expect.

(3) Please also provide the MS/MS spectra for the site-specific occupancy of PenG on MSA-PenG adducts (Fig.S2).

(4) What was the exact concentration of PenG used against HEL and BSA? You specify in the figure that it

was 1:200 per Lys, what is that in terms of concentration?

(5) Please also display the chemical structures of PenV, Benzylpenicillin and PenG and highlight in PenV which part of the molecule is the prodrug part

(6) Please add the number of replicates for the SPR characterization. (Fig. 5e)

Reviewer #3 (Remarks to the Author):

The authors of this manuscript studied the unwanted (B/T cell) immune response to covalent bond-forming drugs, such as penicillin G (PenG) in a mice model. Hypersensitivity and allergic reactions give problems to 8-11% of all patients, however, the prevalence seems to be relatively low.

This study is important as it gives new insights into the binding modes of PenG to carrier proteins and reveals information about present clonotypes.

Generally, there exist many studies identifying and analyzing PenG adducts in human serum and patient samples. Therefore, I wonder, how reliable and significant the herein produced data is. The authors should probably compare their results a bit more in spite of these findings. Moreover, some knowledge exists on serum protein binding of PenG, also this should be compared to existing literature to highlight the novelty of this work.

Further comments are listed in the following:

1. It is commonly accepted that understanding penicillin hypersensitivity has proven to be extremely complex. However, how can the data shown here contribute to the question if an individual might experience a T cell-mediated response? This goes along with the may be very difficult to answer question if free penicillin, ex vivo/in vivo formed PenG-protein adducts play a role for immune response. (Immunogenicity of free PenG was not observed, PenG-specific serological response was not observed)
2. Can the authors translate their results to humans: what is the time of penicillin induced drug hypersensitivity?
3. What is the influence to cytokine expression/cytokine signature?
4. Sites of modification HSA in human serum are already described. Are the findings herein comparable?
5. The authors provide basis to engineer PenG analogues that might induce less hypersensitivity. This is highly interesting and probably the authors can give some structural examples.

Minor:

1. Increase font size, or use bold font in all chemical structures.
2. Figure 1b: please correct the guanidino functional group of arginine.

Reviewer #4 (Remarks to the Author):

Deimel et al provides an interesting look at immune response to haptenated proteins, using a very relevant hapten – penicillin. The resulting complex can activate B-cells and lead to formation of PenG-specific antibodies which are responsible for the induction of penicillin hypersensitivity. Using PenG conjugated to various carrier proteins, they immunize mice and analyzed the antibody repertoire and

mode of binding by performing BCR sequencing and protein NMR, respectively. In the end, solving gaps in the unwanted immune response against PenG could help to develop alternative PenG analogues being less allergenic.

Although interesting, the study falls short on the fact that in real life – haptenization, a likely driving factor for the induction of immune reaction, is likely to occur on a heterogenous sites on different protein. Still, the clinically relevant B cell response can in theory (and shown by the authors) be solely directed to the hapten/penicillin. That response is show here to be restricted to certain V(D)J rearrangements, and as suggested by the authors could be used generate strategies aimed to “germline-guided reverse engineering” without actually doing anything in that direction. In real life, tinkering with the structure of penicillin can be done, might even reduce binding of these antibodies to penicillin while retaining antibiotic activity. However, in the end, this is undoubtedly also immunogenic. In essence, the study points rather to a crucial role of the industry to come up with new antibiotics that have less tendency to form covalent complexes and thus less immunogenic. This is of course long known. In addition, yes, the immune response show a selectiveness towards certain germline sequences, but this is true for any response I have ever seen published, so this is not a surprise.

A minor point would be a few immunological misunderstandings throughout is that conjugation of anything to a self-protein makes it non-self and a subject of immunization. You don't strictly need adjuvants. A good example would be that cellular haptenization by biotinylation or other haptens (including citrullination seen in autoimmunity) is inducing immunity (mostly fading quickly away – but not always). Adjuvants (or infection probably also lowering the threshold for fulminant response) only helps to “push the immunization through to serological maturity”.

Otherwise, the manuscript is extensive and written in a logical order.

Minor points

- Line 92-94, please try to rephrase this sentence for more clarity.
- Line 136, the authors mention ‘close to physiological pH’, but pH11 was used for these conjugation reactions as mentioned in Document S1. Also, only HEL was used for conjugation to PenG, although (line 135) ‘various recombinant carrier proteins’ are mentioned. Please clarify.
- There is no referring to figure S1c.
- Line 139, here, the authors introduce the HEL-PenG conjugate and testing of antisera towards HSA-PenG. However, it is unclear why a PenG conjugate towards an unrelated PenG-modified protein is used to test specificity. Please introduce this section and method better. Would it matter what kind of conjugated proteins are used?
- Figure 1, mention EPT definition also in figure legend to avoid confusion.
- Figure 2, it is not clear what the difference between 2e and 2f is, data are clearly different, but why and what are you showing?
- Line 262, ‘maybe’ or ‘may be’.
- Figure 4b, would have been great to have included HSA alone to see that response, but as this is not required for the main goals of this study, this is not required?
- Sup Doc1, Line 172, looks like a calculation mistake, should be 14974Da, please correct or clarify. Also, line 180 and 193 contain spelling mistakes.

Reviewer #5 (Remarks to the Author):

REVIEWER COMMENTS

Reviewer 1

We appreciate the care with which this reviewer has critically read the review and pointed us in several very helpful directions. We have incorporated most of their suggestions for which we are grateful.

Major points:

1. L203: is alum a good adjuvant to use to mimic the effect of bacterial infection? The authors make the point that innate recruitment is likely to be important in the induction of PenG-specific antibodies (L403). Might a Lipid A analog such as MPLA have been better choice of adjuvant?

We agree that using a TLR-4 agonist might better recapitulate the innate activation profile expected during bacterial infection. However, we have observed in the current study that using LPS (**Fig S3e**) and in other studies MPLA (DOI: 10.1101/2023.06.03.543556) strongly amplifies serum 'stickiness' against hydrophobic targets, such as PenG. Indeed, our unpublished analyses reveal that these adjuvants elicit a polyreactive partly adjuvant-specific antibody response which interferes with assay specificity, and we have therefore chosen to avoid them where possible. Whilst the type of adjuvant will influence the T helper profile and antibody isotype elicited, we consider it highly unlikely that this would influence the specificity of the B cell clonal response to PenG adducts. This is consistent with the point raised by Reviewer 4, that the role of adjuvant in the current study is primarily to push the B cell response to a detectable serological outcome, which our data reflect.

2. L243-263: were the reactivities of the various beta-lactam antibiotics to OVA the same? Could the results to some extent be explained by changes in the distribution of reactions with various residues across OVA?

We agree that antibiotic occupancy on carrier protein might quantitatively influence assay outcome. To interrogate the occupancy of the various B-lactam-modified OVA probes, we measured residual free-amines as described previously (DOI:10.1016/B978-0-12-382239-0.00003-0) as a surrogate for antibiotic occupancy (new Fig. S6; lines 257–258). All probes exhibited a broadly similar reduction in free primary amines compared with the unmodified control, implying similar B-lactam attack of the lysine sidechains for all drugs tested in Fig. 3. Although these data reveal some marginal differences in absolute loading between antibiotics (maximally two-fold; for example, for cephalexin), this cannot account for the 2–3 orders of magnitude difference observed for serum antibody binding in Fig. 3. Moreover, no correspondence with loading variation and response are seen. This outcome is further rationalised by structural and biochemical data outlined in later figures, especially with regard to structural accommodation of the drug nuclei.

3. L372: the recorded binding affinity of FAb MIL-3 to a PenG-adduct chip is comparatively weak, at about 5µM. There was no report of the affinities of MIL-1 and MIL-2- it would have been useful

to include these data, to provide context for the NMR experiments presented in Fig 5. An argument could be made that a solution binding assay would have been more relevant to the NMR and X-ray crystallography studies (eg using fluorescence) although this is a minor point. We thank the reviewer for their suggestion. As demonstrated by the ELISA data (Fig. 4h) all Fabs tested, including MIL-1, MIL-2 and MIL-3, bound specifically to antigen with crudely estimated micromolar affinity. Unfortunately, unlike MIL-3, the MIL-1 and MIL-2 Fabs yielded high background signals when tested by SPR on both dextran and non-dextran-containing chips, precluding determination of accurate K_D values.

We also attempted to derive K_D values from NMR analyses. Similar responses were seen in the uSTA transfer efficiency for MIL1, -2, and -3 indicating similar strength of binding, as shown below in the comparison of the *para* protein transfer efficiencies. The ligand pose was also highly similar in each case, with the majority of the contacts made between the aromatic ring and the protein, in a manner that strongly reflects the crystallography.

MIL-1, MIL-2 and Mil_3 show comparable binding strengths. a) PenG-lys with the *ortho*, *meta* and *para* aromatic protons indicated. b) The proton with the highest uSTA transfer efficiency was the *para*. The specific value was comparable over the three proteins suggesting essentially similar interaction strengths.

Quantitative analysis of the uSTA data suggests a weaker K_D to that observed by SPR. These NMR measurements were necessarily made in the absence of the cryoprotectant glycerol, which in principle could explain differences in apparent binding constants. A detailed analysis of the variation between SPR and the uSTA K_D obtained under different modes and presentation formats is outside the scope of the present work.

L415: in addition to dosing effects and higher drug clearance rates in mice compared to humans, it also needs to be pointed out that there are differences in immunogenic responses between the two.

We agree this is helpful and relevant to flag to the reader. We have appended lines 430–433 to reflect this.

Minor typographical and stylistic changes.

We thank the reviewer for pointing these out—we have amended the manuscript accordingly.

Reviewer 2

We deeply value the thoughtful reflections on the pharmacokinetics. We have subsequently made a series of explanatory amendments to the manuscript to highlight our rationales for the drug dosage and formulations used *in vivo*.

Major points:

1. For the more clinical setup you chose a dose of 2 mg/ml of PenG for pre-incubation and incubated for 16 hrs. Please specify why you chose that concentration? Is it to mimic peak plasma concentrations? In the clinics constant plasma concentrations would only be achieved during infusion. Have you considered varying the incubation time, also including different (lower) concentrations of PenG to see if the effect will remain similar or if some concentrations might not cause the effect?

The reviewer is correct: we used 2 mg/mL to mimic the approximate concentrations maintained by intravenous administration of penicillin in humans (<https://doi.org/10.1093/jac/47.2.129>). We have now incorporated this rationale into the revised manuscript; refer lines 204–205. This experiment was also to demonstrate *ex vivo* that the serum is buffered in a manner that enables adduct formation. Finally, we note that in clinical settings penicillin G is administered passively through an i.v. line for protracted periods of time. We have not titrated the penicillin concentration or incubation time to find a lower threshold for immunogenicity as we believe that the knowledge outcome would not be ethically justified in terms of the substantial additional animal use.

2. Do you have pharmacokinetic information/profiles on PenG, PenV and Benzylpenicillin in your model? You mention a relatively short half-life of PenG. Why did you choose the i.v. route in light of this? The i.m. route gives a different profile. Moreover, there are formulations of PenG which result in longer half-lives after i.v. administration. As your immunogenicity studies are only done with PenG, I would recommend to consider if it was worth to add these data. Moreover, I was wondering which doses for PenG were used and if they were sufficient to achieve the concentrations in plasma you used for your in vitro studies. I am concerned for PenV as you used 2 mg/ml, which does not necessarily translate to 2 mg/ml in vivo. Do you have information if the prodrug PenV is entirely released after intake. Additionally, I am concerned how you make sure that every mouse is getting the same dose to get reproducible results if all animals have access to PenV solution ad libitum. I would appreciate comments on that. In light of that: is there any information available on the PK of Benzylpenicillin after i.m. information with regard to plasma levels? How did you choose the dose of 2 mg?

The existing literature outlines a systematic characterisation of the half-life of PenG and its analogues in various animal models. It is true that the half-life of PenG is relatively short in mice (as outlined in the manuscript introduction and discussion). The estimated half-life in mice is 12–16 mins, and in rabbits is 3–4 h (DOI: [10.1128/AAC.45.4.1078-1085.2001](https://doi.org/10.1128/AAC.45.4.1078-1085.2001)). For our study, we did investigate alternative delivery modes that maintain high circulating drug concentrations, such as use of an osmotic pump to maintain

continuous high drug concentrations (akin to the i.v. lines given to humans), but due to animal experimental ethical constraints we were obliged to use the i.v. dosing schedule as outlined.

As the reviewer alludes to, we tested PenG-Ben as a 'slow-release' formulation of PenG. The dosage of 2mg was again an ethics constraint: it was the highest amount of PenG-Ben we could administer with the limited murine intra-muscular maximum injection volume. However, we note that this amount was nevertheless sufficient to elicit a significant response in some animals.

Regarding the reflections on PenV, we appreciate that the concentration of PenV in water does not necessarily translate to the same concentration *in vivo*. We did not measure the concentration of drug taken up by the animals in this experiment. We maintained animals on PenV for a total of 7 days (split between two separate intervals). Whilst we agree that all mice will not drink an identical amount, it is well accepted that mice consume on average approximately 6 mL of water per day (DOI: 10.1023/a:1020884312053), which translates to approximately 84 mg of PenV. We decided to not gavage the mice; animals showed no signs of dehydration or other welfare concerns, suggesting that the addition of PenV did not affect their water uptake.

3. You provide information on one *S. pneumoniae* strain, which is non-capsulated. Have you tested a capsulated *S. pneumoniae* strain with similar MIC?

The MIC of the *S. pneumoniae* D39 was found to be 0.01 ug/mL, which is equivalent to many capsulated strains of *S. pneumoniae* (refer to Table 1 of Morand & Muhlemann 2007, PMID: 17704255). We would not expect that using a capsulated strain would affect our findings, unless the reviewer has a specific question in mind.

Minor points:

1. It would be more clear to the reader if you specified in Figure 1b which exact moieties react with which part of the protein or exemplify that. You mention it in the introduction and in the initial sentences of the results, but it would be good to also spot it directly in the Figure.

We thank the reviewer for this point—we agree that it would be clearer for the reader to visualise this in Main Figure 1. As suggested, we have moved the lysine–PenG molecular structure diagram from the supplementary to the revised main text to clearly illustrate the product adduct that is formed upon ring-opening reaction of the β -lactam.

2. For the adduct formation in Fig. S1b: please provide MS/MS spectra for the tryptic digest. Moreover, I would encourage to use a schema to show which digestion sites in the protein you actually expect.

As the reviewer will expect, there are many MSMS spectra and rather than select only a few, all raw data have been deposited in PRIDE for direct interrogation (deposition number: PXD052026; Username: reviewer_pxd052026@ebi.ac.uk, Password: Bfh3hgoo). For HEL-PenG we obtained 86% protein coverage,

with 24 peptides identified matching to 284 peptide-spectral matches. For HSA-PenG, we obtained 95% protein coverage, with 115 peptides identified matching to 1758 peptide-spectral matches. As suggested, for HSA-PenG, the main immunogen used throughout this study, we have now included additional analysis as Fig S2b, highlighting the relationship of digestion sites to modification sites that allows an analysis of those frequently vs infrequently modified in a scheme that gives an indication of structural context also. Thank you for this suggestion.

3. Please also provide the MS/MS spectra for the site-specific occupancy of PenG on MSA-PenG adducts (Fig.S2).

We thank the reviewer for this suggestion. We have included MSA-PenG adduct occupancy data acquired via LC-MS/MS in Fig S4.

4. What was the exact concentration of PenG used against HEL and BSA? You specify in the figure that it was 1:200 per Lys, what is that in terms of concentration?

We thank the reviewer for this point of clarification. The exact concentrations used for the modifications are 6.8 and 60 mg/mL for HEL and HSA respectively, where the concentration of protein was 1 mg/mL. This is now mentioned in the revised methods section lines 469-470.

5. Please also display the chemical structures of PenV, Benzylpenicillin and PenG and highlight in PenV which part of the molecule is the prodrug part.

We thank the reviewer for this suggestion of additional structures. We should clarify that PenG and benzylpenicillin are synonymous. Here we use three variants. PenG, PenG-Ben (PenG benzathine) and PenV. We have now included the structure of PenG-Ben in Fig 2g, and additionally included PenV in Fig S3d accordingly, highlighting the phenoxymethyl in PenV rather than phenylacetamide (PenG) sidechain. We note that unlike PenG-Ben, which has an organic counterion (benzathine) designed to lower its solubility and hence dissolution rate, the variation of sidechain PenV rather than PenG is to render the drug more acid-resistant for improved survival in the stomach, which we had outlined in lines 222–225. None are prodrugs *per se* although PenG-Ben's lower dissolution rate allows for a gradual dosing of PenG.

6. Please add the number of replicates for the SPR characterization. (Fig. 5e).

We have now specified this in the associated figure legend; 9 runs at varied concentrations (including duplicates at the lowest concentrations which have been averaged).

Reviewer 3

We thank the reviewer for their thoughtful feedback and compliments on our study. We have taken onboard their suggestions and have made relevant changes to the manuscript accordingly.

Major points:

1. It is commonly accepted that understanding penicillin hypersensitivity has proven to be extremely complex. However, how can the data shown here contribute to the question if an individual might experience a T cell-mediated response? This goes along with the may be very difficult to answer question if free penicillin, ex vivo/in vivo formed PenG-protein adducts play a role for immune response. (Immunogenicity of free PenG was not observed, PenG-specific serological response was not observed).

We thank the reviewer for this probing question. It is true that the immunological underpinnings of penicillin hypersensitivity are heterogenous and can be T cell or antibody-mediated as we outline in the introduction. Generally, antibody mediated hypersensitivity is generally less common but more severe than T cell-based hypersensitivity. Regarding the specific question posed by the reviewer, there are two discrete 'types' of T cell mediated responses to PenG hypersensitivity, for which only one is relevant to the present study. Explicitly pertinent is the concept of 'linked' help wherein a B cell recognises a β -lactam antibiotic-encompassing epitope while a cognate T cell recognises an adjacent peptide within the same antigenic complex. Our data (specifically the immunogenicity of pre-complexed drug-protein conjugates) are consistent with this phenomenon in driving antibody responses. We chose not to investigate direct T cell-mediated hypersensitivity, since this has already been extensively investigated and T cell clones in both mice and humans have been described as recognising penicillin-adducted peptide:MHC complexes. For example, the common broad-spectrum β -lactam piperacillin is highly immunogenic and the T cell-based mechanism is defined: when piperacillin complexes with Lys541 of HSA, the drug-peptide complex is presented and efficiently activates V β 13.1 and V β 17 T cell clones (Meng et al., 2017, PMID: 28438900). In summary, we therefore elected to study antibody-based responses to penicillins because they are more severe and also less well understood than T cell-based responses.

2. Can the authors translate their results to humans: what is the time of penicillin induced drug hypersensitivity?

The timeline from drug administration to hypersensitivity is a challenging topic to tackle. In part, the challenge in testing this is that there are very few people who are 'antibiotic naïve' and thus repeat administration of the drug may be akin to a boost-like response, which happens exceptionally rapidly, as exemplified by the speed with which hypersensitivity can be recalled. However, it is notable that antibody responses against PenG are reported in ~55% of patients treated with the drug, although the disjunct between the rarity of allergy compared to the frequency of anti-PenG responses is likely a function of individual predisposition to atopy linked to differential IgE isotype conversion (Lafaye and Lapresle, 1988, PMID: 3392217).

3. What is the influence to cytokine expression/cytokine signature?

We agree that this is a valid point. We have carried out an investigation of the cytokine expression profile in our HEL/HSA-PenG immunisation model and have now included these data in the revised supplementary figures as new Fig S3. In summary, animals immunised according to the immunisation schedule outlined in Fig 1d and Fig S1c, harvesting the spleens at the terminal timepoint. We observed evidence of HSA-PenG-specific Th responses, as indicated by autologous restimulation and subsequent proliferation (determined by H³ incorporation) and IFN- γ production determined by intracellular cytokine staining. These data are interesting and consistent with the existing literature supporting enhanced immunogenicity of the drug-protein conjugate over the unreacted protein alone

4. Sites of modification HSA in human serum are already described. Are the findings herein comparable?

We thank the reviewer for this—it's true that HSA adduction of β -lactam drugs have previously been shown in the literature, and indeed our data is essentially the same (DOIs: 10.4049/jimmunol.1100647, 10.1124/jpet.111.183871, 10.1021/tx400124m and 10.1371/journal.pone.0090891). We have highlighted the consistency of our data in the manuscript; refer lines 153–154. Our observations are consistent with prior observations of more limited modification sites. Interestingly, certain abundantly modified sites sit close to putative drug binding pockets that have been proposed on the basis of spin-labeling-mediated EPR spectroscopy studies of HSA (DOI: 10.1002/chem.201601810) as well as the so-called Sudlow sites (G. Sudlow, D. J. Birkett, D. N. Wade, *Mol. Pharmacol.* 1975, 11, 824 – 832; G. Sudlow, D. J. Birkett, D. N. Wade, *Mol. Pharmacol.* 1976, 12, 1052 –1061) classically identified by fluorophore displacement.

It might be of interest, as mentioned in response to Reviewer 2, that we have now included additional analysis on the HSA modifications in a structural context in Fig S2b. All raw data is deposited in the PRIDE database.

5. The authors provide basis to engineer PenG analogues that might induce less hypersensitivity. This is highly interesting and probably the authors can give some structural examples.

We agree that the concept of 'reverse engineering' drugs against known allergy-mediating clonotypes is a novel idea. We have not computationally modelled this, but the cross-reactivity ELISA shown in Fig 3 already provides insight into the narrow tolerance in drug nucleus variation (e.g. cephalosporins). When considering how the MIL series engages the thiazolidine (Fig 5f ii.), this outcome is perhaps not surprising. We would note, however, that just because a drug class like cephalosporins might be less likely to engage a MIL-series clonotype, this does not discount the elicitation of its own cognate public clonotype. Thus,

future drug engineering efforts would probably need to be done on a drug-specific and clonotype-specific basis. In the current case, the thiazolidine core would be a target motif to vary; we have elaborated on these points in the discussion. Thank you for this suggestion.

Minor typographical and stylistic changes.

We thank the reviewer for pointing these out—we have amended the manuscript accordingly.

Reviewer 4

We are grateful for the reflections made by the reviewer, and we agree with the general comments made, specifically that pertaining to how drug adduction to an endogenous carrier protein renders the antigenic complex less 'self' than the unmodified endogenous counterpart. Accordingly, we have been explicit about the conceptual/philosophical point in the section using mouse serum albumin, where these ideas are arguably most relevant in our immunization model (refer lines 190–193). We agree that the role of adjuvants is to 'get the B cell response 'over the detectable serological line', so to speak, and our data are absolutely consistent with this.

1. Line 92-94, please try to rephrase this sentence for more clarity.

Thank you; we have amended.

2. Line 136, the authors mention 'close to physiological pH', but pH11 was used for these conjugation reactions as mentioned in Document S1. Also, only HEL was used for conjugation to PenG, although (line 135) 'various recombinant carrier proteins' are mentioned. Please clarify.

We apologise for this confusion; the manuscript that we adapted the modification protocol from (Padovan et al 1997, PMID: 9209477) used 0.1 M Na₂CO₃, pH 11. We, however, tested a range of pH (as outlined in the Document S1 lines 126–131; Figure S1). And, as outlined in the Methods section (lines 467), we chose the final pH of 8. Again, we apologise for the discrepancy and confirm that all *ex vivo* single protein modifications used in immunisation experiments were conducted at pH = 8. We have amended the Document S1 to reflect this.

3. There is no referring to figure S1c.

We have subsequently moved this figure to Fig 1 and now directly refer to it in the revised manuscript.

4. Line 139, here, the authors introduce the HEL-PenG conjugate and testing of antisera towards HSA-PenG. However, it is unclear why a PenG conjugate towards an unrelated PenG-modified

protein is used to test specificity. Please introduce this section and method better. Would it matter what kind of conjugated proteins are used?

We thank the reviewer for bringing up this important point and have now included a rationale for this approach in the revised results text (lines 143–146). In short, this method is akin to measuring classical hapten-carrier-type responses such as against nitrophenol: by testing the serological reactivity of a similarly modified unrelated carrier protein, reactivity is specifically screened against the common chemical motif (in this case, the PenG adduct). In theory it might be possible to screen antibody binding to drug directly coupled to a solid phase (eg. some form of modified support or ELISA plate), however these substrates can themselves confound in their often higher background binding and so complexing of the drug to a lysine on a different protein presents it in a state closer and so more comparable to the original immunogens used here as well as to the presentation in a clinical setting, which we consider to be more biologically relevant.

5. Figure 1, mention EPT definition also in figure legend to avoid confusion.

Have amended accordingly, thank you.

6. Figure 2, it is not clear what the difference between 2e and 2f is, data are clearly different, but why and what are you showing?

We apologise—this is a typographical error in the axis title. Although it is clear from the results text, panel 2f is the reactivity against the unmodified protein backbone (OVA), while 2e is against OVA-PenG. The data reflect reactivity against the adduct specifically. We have corrected the axis title and thank the reviewer for pointing this out.

7. Line 262, ‘maybe’ or ‘may be’.

Thank you, we have fixed this.

8. Figure 4b, would have been great to have included HSA alone to see that response, but as this is not required for the main goals of this study, this is not required?

We did not include a molecular probe to evaluate B cells against the protein backbone. As the reviewer correctly acknowledges, this was not the goal of this experiment and we therefore focused on isolating adduct-specific clones.

9. Sup Doc1, Line 172, looks like a calculation mistake, should be 14974Da, please correct or clarify. Also, line 180 and 193 contain spelling mistakes.

The mass was a typographical error that has been correct, along with that in lines 180 and 193. We thank the reviewer for bringing these to our attention.

REVIEWERS' COMMENTS

Reviewer #1 (Remarks to the Author):

The authors have responded well to the points raised in my original review; the manuscript has been revised appropriately and I have no further criticisms to raise.

Reviewer #2 (Remarks to the Author):

I would like to thank the authors for answering my questions thoroughly and for providing amendments and additional data/schemes. I have no further comments.

Reviewer #3 (Remarks to the Author):

The authors have carefully worked on the manuscript and answered all my questions satisfactorily.